# Relationships between objective structured clinical examination, computer-based testing, and clinical clerkship performance in Japanese medical students

**Nobuyasu Komasawa** **\*, Fumio Terasaki, Takashi Nakano, Ryo Kawata**

Medical Education Center, Osaka Medical College, Takatsuki, Japan

\* ane078@osaka-med.ac.jp

## Abstract

### Background

It is unclear how comprehensive evaluations conducted prior to clinical clerkships (CC), such as the objective structured clinical examination (OSCE) and computer-based testing (CBT), reflect the performance of medical students in CC. Here we retrospectively analyzed correlations between OSCE and CBT scores and CC performance.

### Methods

Ethical approval was obtained from our institutional review board. We analyzed correlations between OSCE and CBT scores and CC performance in 94 medical students who took the OSCE and CBT in 2017 when they were 4th year students, and who participated in the basic CC in 2018 when they were 5th year students.

### Results

Total scores for OSCE and CBT were significantly correlated with CC performance (P<0.001, each). More specifically, medical interview and chest examination components of the OSCE were significantly correlated with CC performance (P = 0.001, each), while the remaining five components of the OSCE were not.

### Conclusion

Our findings suggest that the OSCE and CBT play important roles in predicting CC performance in Japanese medical education context. Among OSCE components, medical interview and chest examination were suggested to be important for predicting CC performance.

**Funding:** The authors have no affiliation with any manufacturer of any device described in the manuscript and declare no financial interest in relation to the material described in the manuscript. Financial support for the study was provided by Osaka Medical College which had no role in study design, data collection and analysis, publication decisions, or manuscript preparation.

**Competing interests:** The authors have declared that no competing interests exist.

## Introduction

Seamless medical education in which students gradually acquire professional abilities from when they are undergraduates up until they become postgraduates is important from the perspective of outcome-based education [1]. To achieve this goal, effective clinical training methods are needed which allow for a smooth transition from undergraduate medical education to basic skill acquisition as a postgraduate [2].

In Japan, clinical clerkships (CC)s form the basis of clinical training. In contrast to conventional clinical training, which involves only observation and no practice, CC have students participate as members of a medical team to perform actual medical procedures and care. The range of medical procedures allowed to be performed by students is defined and carried out under the supervision of an instructing doctor [3]. This enables students to acquire practical clinical skills. In this regard, students are required to have a sense of identity and personal responsibility [4]. Clinical training throughout the various departments of a hospital is carried out in the form of CCs which are driven by curricula for diagnoses and treatments [5].

Assuring basic clinical competency in medical students prior to CC is essential from a medical safety perspective. In order to validate the basic clinical competency of medical students, the objective structured clinical examination (OSCE) and computer-based testing (CBT) were introduced in 2005 as standardized tests, organized by the Common Achievement Tests Organization (CATO) [http://www.cato.umin.jp/], to be taken by medical students. The OSCE evaluates clinical skills and communication skills using simulated patients and simulators [6] [7], and the CBT basic clinical knowledge. The OSCE and CBT are mandatory for 4th year students in Japanese medical schools. Medical students are recognized by the Association of Japanese Medical Colleges as "student doctor" once they pass both examinations. After this certification, medical students can participate in the CC.

Although previous studies examined the CC performance by mini-clinical evaluation exercise (mini-CEX) [8][9], the relation between OSCE or CBT and CC performance has not been fully validated. Furthermore, no study has evaluated which skill components measured in the OSCE reflect student performance in CCs in Japanese medical education context.

Thus, we decided to evaluate the relationships between OSCE components or CBT and CC performance in Japanese medical education contexts. Accordingly, the present study aimed to retrospectively analyze correlations of various components of the OSCE and CBT with CC performance.

## Material and methods

### Ethical considerations

This study was approved by the Research Ethics Committee of Osaka Medical College (No.2806). All data were fully anonymized before accessing them and our research ethics committee waived the requirement for informed consent.

### Study population

As with other medical schools in Japan, medical students of Osaka Medical College take the OSCE and CBT in their 4th year, and participate in CCs in their 5th and 6th years. The students have undergone all basic and clinical medicine lectures and skill training utilizing simulation before OSCE and CBT. Once they complete their CCs, medical students then take the graduation examination. From 2020, a post-CC OSCE will be introduced by CATO formally to evaluate clinical skills cultivated during CCs (Fig 1).

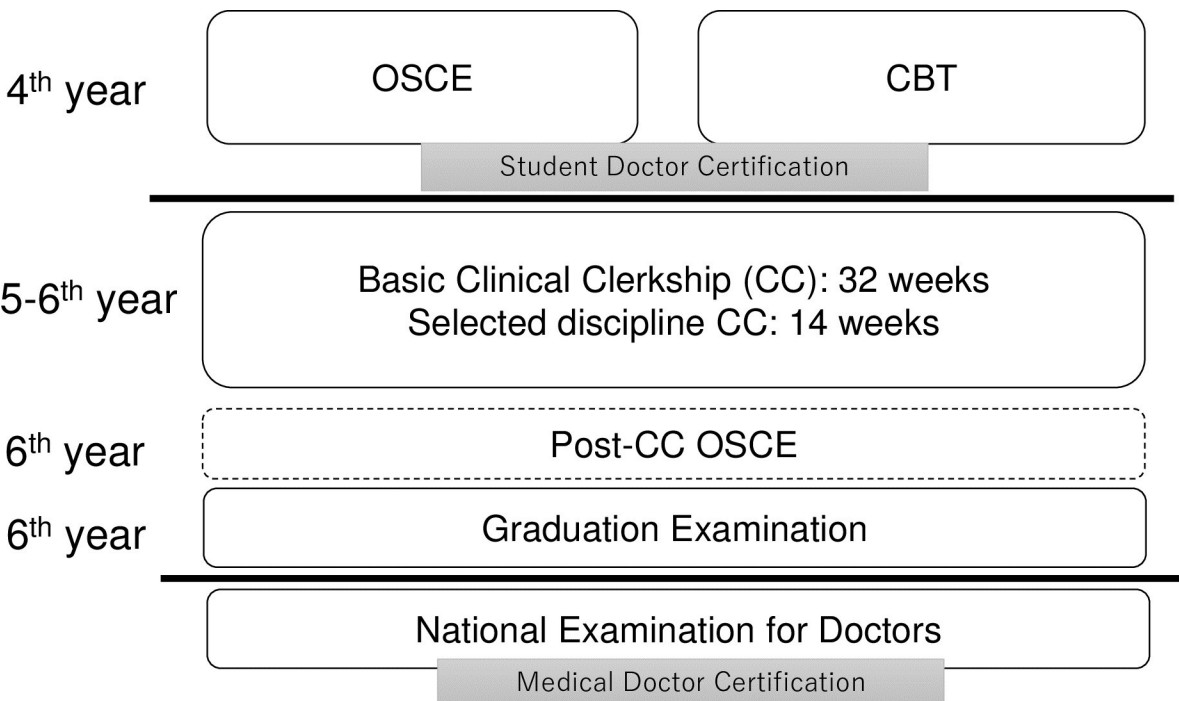

**Fig 1. Schematic summarizing relationships between objective structured clinical examination (OSCE), computer-based testing (CBT), and clinical clerkships (CCs) at Osaka Medical College.**

Subjects of the present study were medical students of Osaka Medical College who were 4th year students in 2017 and 5th year student in 2018. We excluded students who did not advance to 5th year status in 2018.

### Study measures

**OSCE content and evaluation.**    The OSCE evaluates various aspects of clinical competency. The OSCE included the following seven themes: medial interview, head and neck examination, chest examination, abdominal examination, neurological examination, emergency response, and basic clinical technique. The OSCE is carried out in seven stations, with one station dedicated to a 10-min medical interview, and the remaining six stations to physical examinations and basic skills in 5-min for each. In the 5 or 10 minutes, students perform core clinical skills such as medical interview and physical examination [10].

In the present study, student performance was evaluated by two examiners using a checklist. Scores on each component of the OSCE was based on the average of the scores assigned by the two examiners. Examiners evaluate the communication, medical safety, and consultation skills accordingly on the checklist. The examiners underwent about three hours evaluation training for standardization based on common text and video provided by CATO. Each student take examination in the all seven skill stations and total score was calculated by the average of seven skills. The examination also strictly checks the identification of students by validating their names and ID numbers. Examiners from other universities are routinely invited to validate internal evaluations during the OSCE.

**CBT content and evaluation.**    The CBT consists of multiple-choice questions and extended matching items, and students are required to answer 320 questions about basic clinical knowledge over the course of six hours. Evaluation was performed by the 240 questions

which the difficulty and discriminating power was validated from the past pooling data. The remaining 80 questions were trial questions which are not used for the evaluation. The questions are standard tested by the CATO. The CBT includes clinical disciplines and related basic medicine knowledges. Scores for the CBT are machine-calculated and scoring rate was evaluated.

**Clinical clerkship (CC) content and evaluation.** Medical students participate in a basic CC during their 5th year. The basic CC involves participation in CCs of all clinical departments of the hospital over the course of 32 weeks, with each CC spanning about one to two weeks in duration. Once students complete the basic CC, they must then select a discipline they wish to study for 14 weeks (Figure).

During the CCs, supervising (teaching) doctors of each department evaluate the clinical skills of students using an evaluation sheet based on the mini- CEX and Direct Observation of Procedural Skills (DOPS) [11][12]. Accomplishment consists of 5-point evaluation sheet for 16 parts (80%), subjective evaluation by the organizer of each department (10%), and written report (10%) (Fig 2).

Scores for each CC are collected by the medical education center and are used to calculate an average score. In this study, we used the basic CC (32 weeks) score, since all medical students are required to participate in the basic CC.

## Statistical analysis

Statistical analyses were performed using JMP$^{®}$ 11 (SAS Institute Inc., Cary, NC, USA). Results were compared using Pearson's correlation test. Data are presented as mean ± SD. $P < 0.05$ was considered statistically significant.

## Results

We analyzed scores of 94 medical students who participated in the OSCE and CBT in 2017, and the basic CC in 2018. As shown in Table 1, medical students generally achieved scores ranging from 80%-90% for the OSCE, CBT, and CC.

### Correlations of OSCE and CBT scores with CC scores

Correlations of OSCE and CBT scores with CC scores are shown in Table 2. Total scores for OSCE and CBT were significantly correlated with CC scores (P<0.001 each). When analyzed by OSCE components, medical interview and chest examination scores were significantly correlated with CC scores (P = 0.001, each), while the remaining five component scores were not.

### Correlations between OSCE and CBT scores

We evaluated correlations between OSCE and CBT scores, and found no significant correlations between them (Table 3). There also were no significant correlations between each OSCE component score and CBT score (Table 3).

## Discussion

Our study showed that total scores for OSCE and CBT were significantly correlated with CC scores. This suggest that OSCE and CBT can be an effective indicator of CC performance in Japanese medical education. From the specific correlation analysis, medical interview and chest examination scores were significantly correlated with CC scores.

Physical examinations and medical interviews are essential skills and being able to evaluate information from these provides information important for diagnosis and treatment during

**80 %**: Evaluation 16 parts of knowledge, skill, and attitude
utilizing 5-point scale evaluation sheet
(1 Insufficient, 2 Sufficient, 3 Satisfying, 4 good, 5 very good)
**Knowledge (4 parts)**
Amount, Interpretation, Differential Diagnosis, Problem Solution
**Skill (7 parts)**
Communication, Physical examination, Basic inspection, Basic technique,
Planning, Information gathering, Documentation, Presentation
**Attitude (4 parts)**
Attire and courtesy, Attitude to patients, Positiveness, Cooperativeness

**10 %**: Subjective evaluation by the organizer of each department

**10 %**: Evaluation of written reports by students

**Fig 2. Contents of clinical clerkships (CC) evaluation in our college.** Accomplishment consists of 5-point evaluation sheet for 16 parts (80%), subjective evaluation by the organizer (10%), and written report (10%).

CCs [13][14]. In clinical settings, it is not rare to overlook physical findings or perform evaluations incorrectly. Incorrect assessment of physical findings can lead to errors in diagnosis, which may result in an adverse outcome for the patient [15][16]. Accordingly, from the perspectives of clinical competency and outcome-based education, assuring the quality of both technical and non-technical skills of medical students before CCs is essential.

In the present study, total scores for OSCE and CBT showed strong and significant correlations with CC performance, as reflected in CC scores. These data validate the OSCE and CBT as measures to assure competency prior to participating in CCs. Interestingly, no components of the OSCE were significantly correlated with total CBT score. This suggests that competency as evaluated using the OSCE and CBT are different, and a combination of both could provide a better sense of the competency of medical students prior to CCs.

When OSCE components were considered individually, medical interview and chest examination components were significantly correlated with CC performance, while the remaining five components were not. One potential explanation for this is that, of the seven components of the OSCE, medical interviews and chest examinations are performed the most often during CCs. Thus, focusing training on these skills may contribute to better CC performance [17][18].

**Table 1. Scores for objective structured clinical examination (OSCE), computer-based testing (CBT), and clinical clerkship (CC).**

|  | Medical interview | Head and neck examination | Chest examination | Abdominal examination | Neurological examination | Emergency response | Basic Technique | Total OSCE score | Computer-based testing | Clinical Clerkship |
|---|---|---|---|---|---|---|---|---|---|---|
| **Average** | 80.3 | 92.2 | 81.8 | 94.3 | 90.8 | 90.6 | 82.4 | 87.5 | 80.1 | 78.0 |
| **SD** | 9.9 | 6.6 | 10.8 | 4.9 | 7.6 | 6.6 | 5.8 | 3.9 | 7.5 | 2.1 |

**Table 2. Correlations of objective structured clinical examination (OSCE) and computer-based testing scores with clinical clerkship (CC) scores.**

|  | Medical interview | Head and neck examination | Chest examination | Abdominal examination | Neurological examination | Emergency response | Basic Technique | Total OSCE score | Computer-based testing |
|---|---|---|---|---|---|---|---|---|---|
| **R** | 0 | 0.059 | 0.329 | 0.166 | 0.158 | 0.124 | 0.161 | 0.377 | 0.346 |
| **Co-efficient** | 0.075 | 0.004 | 0.108 | 0.028 | 0.025 | 0.015 | 0.026 | 0.141 | 0.120 |
| **P** | 0.001* | 0.570 | 0.001* | 0.109 | 0.130 | 0.234 | 0.122 | <0.001* | <0.001* |

*$P < 0.05$.

In contrast, components other than medical interviews and chest examinations were not significantly correlated with CC performance. One possible reason for this is the lack of opportunities to use such skills. For example, medical students are not permitted to perform emergency response such as advanced life support alone [19][20]. As medical students should acquire these basic clinical competencies after medical doctor certification, some educational method for compensating the gap is warranted [21].

To overcome this problem, we believe that simulation-based education (SBE) can be a powerful solution to compensate for the lack of opportunities to exercise these skills [22]. As SBE methods have been developed and are widely used to acquire both technical and non-technical skills in medical education [23][24], combination of SBE and CC could potentially maximize the competency of medical students. For example, medical students can rephrase the resuscitation utilizing simulator which they watched in the emergency ward. Such combination can enhance the CC performance in medical students.

Medical educators are expected to improve CC program by including SBE method to compensate low-frequent clinical skills [25]. They can also utilize SBE for formative assessment to improve teaching and learning in clinical region.

This study has a number of limitations worth noting. First, as data were obtained from a single institution, our findings may not be generalizable to other medical schools [26][27]. However, our results likely apply to medical schools in Japan given the core medical curriculum adopted throughout the country. Second, the CBT scores are generally high and small variation, this might have caused some bias for correlation analysis. Third, we excluded the students who did not progress to perform correlation analysis more accurately, as we considered that content of CC may differ year by year. However, this may have caused bias. Fourth, we evaluated the total OSCE score by the average of seven stations which may have lacked statistical justification. Lastly, we evaluated the overall score for CC. Correlations between the OSCE and CBT and the various aspects of CC may provide further insight into how these test instruments relate to actual medical practice by medical students. In this regard, it will be interesting to evaluate the relationship between CC performance and post-CC OSCE scores once the post-OSCE is implemented in the 2020 curriculum year.

**Table 3. Correlations between scores of objective structured clinical examination (OSCE) components and computer-based testing (CBT).**

|  | Medical interview | Head and neck examination | Chest examination | Abdominal examination | Neurological examination | Emergency response | Basic Technique | Total OSCE score |
|---|---|---|---|---|---|---|---|---|
| **R** | -0.045 | 0.0618 | 0.179 | -0.021 | -0.059 | 0.044 | 0.04 | 0.063 |
| **Co-efficient** | 0.021 | 0.004 | 0.032 | 0.0005 | 0.003 | 0.002 | 0.002 | 0.004 |
| **P** | 0.667 | 0.560 | 0.085 | 0.841 | 0.579 | 0.672 | 0.702 | 0.549 |

In conclusion, our findings suggest that the OSCE and CBT play important roles in predicting CC performance. In particular, medical interview and chest examination components of the OSCE were particularly relevant for predicting CC performance.

## Supporting information

**S1 Data.**
(XLSX)

## Author Contributions

**Conceptualization:** Nobuyasu Komasawa, Takashi Nakano, Ryo Kawata.

**Methodology:** Nobuyasu Komasawa.

**Supervision:** Fumio Terasaki, Takashi Nakano, Ryo Kawata.

**Writing – original draft:** Nobuyasu Komasawa.

**Writing – review & editing:** Fumio Terasaki, Takashi Nakano, Ryo Kawata.

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
