## [Decision Letter · Decision Letter 0]

20 Jan 2020

PONE-D-19-29629

Relationships between Objective Structured Clinical Examination, Computer-based Testing, and Clinical Clerkship Performance in Japanese Medical Students

PLOS ONE

Dear Dr Komasawa,

Thank you for submitting your manuscript to PLOS ONE. After careful consideration, we feel that it has merit but does not fully meet PLOS ONE’s publication criteria as it currently stands. Therefore, we invite you to submit a revised version of the manuscript that addresses the points raised during the review process.

We would appreciate receiving your revised manuscript by Mar 05 2020 11:59PM. To enhance the reproducibility of your results, we recommend that if applicable you deposit your laboratory protocols in protocols.io, where a protocol can be assigned its own identifier (DOI) such that it can be cited independently in the future. For instructions see: http://journals.plos.org/plosone/s/submission-guidelines#loc-laboratory-protocols

We look forward to receiving your revised manuscript.

Kind regards,

Conor Gilligan

Academic Editor

PLOS ONE

Additional Editor Comments (if provided):

This paper addresses an important topic in medical education but some further detail and justification is needed to strengthen the paper and ensure it makes a valuable contribution. Both reviewers make helpful points which you should take into consideration in preparing a revised version. In particular, I agree that the study needs clearer justification and I would think that further discussion of the theories associated with competency-based assessment and the challenges of teaching and assessment in clinical settings would be an important addition. Also, please provide further detail on the methods including the assessment measures and scoring systems. The discussion will need to change to reflect these changes to other sections.

Journal Requirements:

2. In ethics statement in the manuscript and in the online submission form, please provide additional information about the records used in your retrospective study. Specifically, please ensure that you have discussed whether all data were fully anonymized before you accessed them and/or whether the IRB or ethics committee waived the requirement for informed consent. If students provided informed written consent to have data from their medical records used in research, please include this information.

3. Thank you for stating the following in the Declaration of Interests Section of your manuscript:

"Financial support for the study was provided by Osaka Medical College which had no role in study design,

data collection and analysis, publication decisions, or manuscript preparation."

Reviewers' comments:

Reviewer's Responses to Questions

**Comments to the Author**

1. Is the manuscript technically sound, and do the data support the conclusions?

Reviewer #1: Yes

Reviewer #2: Yes

2. Has the statistical analysis been performed appropriately and rigorously? 

Reviewer #1: Yes

Reviewer #2: Yes

3. Have the authors made all data underlying the findings in their manuscript fully available?

Reviewer #1: Yes

Reviewer #2: Yes

4. Is the manuscript presented in an intelligible fashion and written in standard English?

Reviewer #1: Yes

Reviewer #2: Yes

5. Review Comments to the Author

Reviewer #1: Thank you for the opportunity to review this paper. It is an interesting piece of work, however, there are issues that need to be addressed and clarified by the authors in different parts of the manuscript.

Introduction

The introduction lacks in depth discussion and information about the relationship between OSCE, written examination and Mini-CEX. The authors have provided the organizational context in the introduction but have failed to highlight what is already known in the literature about the relationship between OSCEs, written examination and Mini-CEX. Some examples of published literature include

• Susan Humphrey-Murto, Mylène Côté, Debra Pugh & Timothy J. Wood (2018) Assessing the Validity of a Multidisciplinary Mini-Clinical Evaluation Exercise, Teaching and Learning in Medicine, 30:2, 152-161, DOI: 10.1080/10401334.2017.1387553

• Rogausch A, Beyeler C, Montagne S, Jucker-Kupper P, Berendonk C, Huwendiek S, Gemperli A, Himmel W. The influence of students’ prior clinical skills and context characteristics on mini-CEX scores in clerkships–a multilevel analysis. BMC medical education. 2015 Dec;15(1):208.

Given that there are published evidence highlighting the relationship between the above variables, the authors need to identify the gap and state the justification for the study.

In addition, some of the references listed such as references, one (1) to four (4) do not reflect the statements written by the authors in the first paragraph. Please use appropriate references.

Methods

Study measures

OSCE content and evaluation

The authors stated that seven aspects were covered in the OSCE. Could the authors state the clinical disciplines that were examined? Was it only one discipline or was the OSCE conducted in all disciplines (Internal Medicine, Pediatrics, Obs and Gyn, Surgery, Family medicine, Emergency, and Anesthesiology)?

How was the overall score calculated?

CBT Content and Evaluation

What content was covered? Were the clinical disciplines assessed? If so, please state it.

Clinical clerkship content and evaluation

The authors have stated that an evaluation sheet based on Mini-CEX and DOPS was used. Could the authors include a copy of the evaluation sheet?

In addition, the authors have stated that the basic clinical clerkship was conducted across all disciplines. How was the average and overall score for the mini-CEX calculated as well as the scores for the different clinical disciplines? Was the form adapted for the different clinical disciplines? If so, please provide the details

Statistical analysis

The statistical analysis should be updated based on the information provided above related to the methods.

Results

As stated in the methods section, the authors need to provide further information on how the average score reported were calculated for the OSCE, CBT and CC. This applies to all sections of the results. Given the above issues, the results reported by the authors cannot be verified.

Discussion

Given the concerns raised about the methodology and results, the discussion needs to be re-written to align with the updated information.

However, the first paragraph in the discussion is a repetition of the first paragraph in the introduction. The authors need to consider how to present the findings of the study in relation to existing evidence.

Other issues

Some references listed in text do not reflect the statements written by the authors. It is important for the authors to use appropriate references.

Reviewer #2: This manuscript is very well written. It looks at the relationship between objective structured clinical exam, computer-based testing and clinical clerkship performance in Japanese medical student. The transition from medical student to practising doctor is a very topical issue in medical education at the moment and hence this manuscript is very welcomed.

Well, the paper has been written very well with good statistical rigour, however, there are a few clarifications that is needed.

1. It would be good to know what teaching was done prior to the test. Did all 94 students attend the session. was it mandatory? was it right before the test or was there a lag period? This was not clear on the manuscript and would have a significant impact on the results. Please clarify.

2. What elements of the examination (head and neck exam; neurological examination etc.) was expected in the five minute OSCE period. So, specifically for the chest exam, was it just the preacordium that was examined in the five minutes? For neurological exam was it all aspects of sensation, proprioception, reflexes for upper and lower limb that was done in the five minutes? Please clarify.

3. Did the 2 examiners scores correlate? An average was achieved. Was examiner training provided to ensure consistency?

4. Typo in paragraph 2 of study measures. 'the examination also strictly checks the identification of students.....'

5. For the computer based testing, were the 320 question standard tested beforehand? If so, was this to the standard of a fourth-year or a 5th year medical student?

6. In addition clarification should be provided as to what aspects of clinical knowledge was tested. Was it all subjects or just acute medicine and acute surgery that is applied during clinical clerkship?

7. It looks like the mini clinical evaluation exercise and the direct observation of procedural skills assessement tools were graded for correlation purposes. Could the grading/marksheets be provided, please. Traditionally these assessments are used for feedback purposes and not usually graded.

Many thanks.

6. PLOS authors have the option to publish the peer review history of their article (what does this mean?). If published, this will include your full peer review and any attached files.

Reviewer #1: No

Reviewer #2: Yes: Gozie Offiah

---

## [Author Response · Author response to Decision Letter 0]

13 Feb 2020

To the Editor (Prof. Conor Gilligan, M.D.)

Editor comments: This paper addresses an important topic in medical education but some further detail and justification is needed to strengthen the paper and ensure it makes a valuable contribution. Both reviewers make helpful points which you should take into consideration in preparing a revised version. In particular, I agree that the study needs clearer justification and I would think that further discussion of the theories associated with competency-based assessment and the challenges of teaching and assessment in clinical settings would be an important addition. Also, please provide further detail on the methods including the assessment measures and scoring systems. The discussion will need to change to reflect these changes to other sections.

Thank you for appreciating the clinical importance regarding our article.

We revised the manuscript faithfully according to the insightful comments of the reviewers as possible, especially for justification and discussion on the theory on competency-based assessment and challenges of teaching and assessment. We believe that simulation-based education can be a solution for competency-based assessment and dissolving the problems associated with teaching and learning in clinical settings.

We believe that the quality of our article improved significantly by their valuable comments.

To Reviewer 1

Thank you for appreciating the educational significance of our article.

1. Reviewer comments; Introduction

The introduction lacks in depth discussion and information about the relationship between OSCE, written examination and Mini-CEX. The authors have provided the organizational context in the introduction but have failed to highlight what is already known in the literature about the relationship between OSCEs, written examination and Mini-CEX. Some examples of published literature include

• Susan Humphrey-Murto, Mylène Côté, Debra Pugh & Timothy J. Wood (2018) Assessing the Validity of a Multidisciplinary Mini-Clinical Evaluation Exercise, Teaching and Learning in Medicine, 30:2, 152-161, DOI: 10.1080/10401334.2017.1387553

• Rogausch A, Beyeler C, Montagne S, Jucker-Kupper P, Berendonk C, Huwendiek S, Gemperli A, Himmel W. The influence of students’ prior clinical skills and context characteristics on mini-CEX scores in clerkships–a multilevel analysis. BMC medical education. 2015 Dec;15(1):208.

Given that there are published evidence highlighting the relationship between the above variables, the authors need to identify the gap and state the justification for the study.

In addition, some of the references listed such as references, one (1) to four (4) do not reflect the statements written by the authors in the first paragraph. Please use appropriate references. 

Thank you for constructive comments regarding our article. According to the suggestion of Reviewer#1, we re-wrote the introduction part for justification. We also changed the references to appropriate ones.

2. Reviewer comments; Methods

Study measures

OSCE content and evaluation

The authors stated that seven aspects were covered in the OSCE. Could the authors state the clinical disciplines that were examined? Was it only one discipline or was the OSCE conducted in all disciplines (Internal Medicine, Pediatrics, Obs and Gyn, Surgery, Family medicine, Emergency, and Anesthesiology)?

How was the overall score calculated?

CBT Content and Evaluation

What content was covered? Were the clinical disciplines assessed? If so, please state it.

Clinical clerkship content and evaluation

The authors have stated that an evaluation sheet based on Mini-CEX and DOPS was used. Could the authors include a copy of the evaluation sheet?

In addition, the authors have stated that the basic clinical clerkship was conducted across all disciplines. How was the average and overall score for the mini-CEX calculated as well as the scores for the different clinical disciplines? Was the form adapted for the different clinical disciplines? If so, please provide the details

Statistical analysis

The statistical analysis should be updated based on the information provided above related to the methods.

Thank you for constructive comments regarding our article. According to the suggestion of Reviewer#1, we changed our manuscript accordingly.

1, We apologize for the expression. We performed this basic OSCE which includes basic OSCE which is common to all disciples. The seven parts is chest examination, abdominal examination, head and neck examination, neurological examination, emergency response, basic technique which we evaluated in the OSCE. We added this information clearly.

2, Over all score was calculated of the average of the seven OSCE stations.

3, CBT included basic clinical disciplines and basic medicine knowledges. 

4, We also added the summary of clinical clerkship evaluation sheet as Figure 2. We made this sheet which we can make scoring and evaluation definitely.

3, Reviewer comments; Results

As stated in the methods section, the authors need to provide further information on how the average score reported were calculated for the OSCE, CBT and CC. This applies to all sections of the results. Given the above issues, the results reported by the authors cannot be verified.

Thank you for constructive comments regarding our article. According to the suggestion of Reviewer#1, we clarified the average score calculation method. We again checked our statistics method and confirmed the correct results. 

4, Reviewer comments; Discussion

Given the concerns raised about the methodology and results, the discussion needs to be re-written to align with the updated information.

However, the first paragraph in the discussion is a repetition of the first paragraph in the introduction. The authors need to consider how to present the findings of the study in relation to existing evidence.

Thank you for constructive comments regarding our article. According to the suggestion of Reviewer#1, we re-wrote the discussion part. 

We sincerely appreciate for your insightful suggestions regarding our article. We believe that our manuscript has been significantly improved by your valuable comments.

To Reviewer 2

Thank you very much for appreciating the educational significance of our article. 

We all agree with all your insightful comments and revised the manuscript accordingly.

1. Reviewer comment; It would be good to know what teaching was done prior to the test. Did all 94 students attend the session. was it mandatory? was it right before the test or was there a lag period? This was not clear on the manuscript and would have a significant impact on the results. Please clarify.

Thank you for constructive comments regarding our article. According to the suggestion of Reviewer#2, we added the concise information of the OSCE evaluation in the introduction and method part.

2. Reviewer comment; What elements of the examination (head and neck exam; neurological examination etc.) was expected in the five minute OSCE period. So, specifically for the chest exam, was it just the preacordium that was examined in the five minutes? For neurological exam was it all aspects of sensation, proprioception, reflexes for upper and lower limb that was done in the five minutes? Please clarify.

Thank you for constructive comments regarding our article. According to the suggestion of Reviewer#2, we added more information about the OSCE. Actually, the scenario theme was randomly selected and medical students perform core examination skill in each part. We added this in the method part.

3. Reviewer comment; Did the 2 examiners scores correlate? An average was achieved. Was examiner training provided to ensure consistency?

Thank you for constructive comments regarding our article. We confirmed that 2 examiners scores are correlated. Also, we perform 3 hour evaluation training before OSCE. According to the suggestion of Reviewer#2, we added some training for OSCE evaluation.

4. Reviewer comment; Typo in paragraph 2 of study measures. 'the examination also strictly checks the identification of students.....'

Thank you for constructive comments regarding our article. According to the suggestion of Reviewer#2, we corrected the expression.

5. Reviewer comment; For the computer based testing, were the 320 question standard tested beforehand? If so, was this to the standard of a fourth-year or a 5th year medical student?

Thank you for constructive comments regarding our article. This is a standard for 4th-year student. According to the suggestion of Reviewer#2, we added this in the introduction part.

6. Reviewer comment; In addition clarification should be provided as to what aspects of clinical knowledge was tested. Was it all subjects or just acute medicine and acute surgery that is applied during clinical clerkship?

Thank you for constructive comments regarding our article. The CBT includes basic clinical knowledge in all disciplines. According to the suggestion of Reviewer#2, we added this information in the method part.

7. Reviewer comment; It looks like the mini clinical evaluation exercise and the direct observation of procedural skills assessement tools were graded for correlation purposes. Could the grading/marksheets be provided, please. Traditionally these assessments are used for feedback purposes and not usually graded.

Thank you for constructive comments regarding our article. According to the suggestion of Reviewer#2, we added the clinical clerkship grading sheet as Figure 2. However, OSCE evaluation sheer contain some restrictions by Common Achievement Test Organization in Japan. We added the evaluation standards of OSCE in the methods part as possible with citation.

We sincerely appreciate for your insightful suggestions regarding our article. We believe that our manuscript has been significantly improved by your valuable comments.

---

## [Decision Letter · Decision Letter 1]

2 Mar 2020

PONE-D-19-29629R1

Relationships between Objective Structured Clinical Examination, Computer-based Testing, and Clinical Clerkship Performance in Japanese Medical Students

PLOS ONE

Dear Dr Komasawa,

Thank you for submitting your manuscript to PLOS ONE. After careful consideration, we feel that it has merit but does not fully meet PLOS ONE’s publication criteria as it currently stands. Therefore, we invite you to submit a revised version of the manuscript that addresses the points raised during the review process.

We would appreciate receiving your revised manuscript by Apr 16 2020 11:59PM. To enhance the reproducibility of your results, we recommend that if applicable you deposit your laboratory protocols in protocols.io, where a protocol can be assigned its own identifier (DOI) such that it can be cited independently in the future. For instructions see: http://journals.plos.org/plosone/s/submission-guidelines#loc-laboratory-protocols

We look forward to receiving your revised manuscript.

Kind regards,

Conor Gilligan

Academic Editor

PLOS ONE

Additional Editor Comments (if provided):

The authors have attempted to address the reviewers comments but in many cases have done so superficially, resulting in a paper that continues to lack clarity.

1. The conclusion section of the abstract needs to make a stronger ‘so what?’ point.

2. Reviewer 1 suggested a deeper exploration of the topic in the introduction and I do not feel that this has been addressed.

3. On page 3 – ‘assuring basic clinical competency is essential’ – are the authors referring to achieving this prior top CC?

4. On page 4, the authors have added some detail but the writing. Is unclear and includes repetition. The paragraph starting ‘The OSCE…’ needs to be revised.

5. Page 5 please delete ‘We have discussed that’ and begin this sentence with ‘All data’

6. Why were students who did not progress excluded? Might this have introduced bias? Also, I would expect that this group might increase the variance in the findings which would be statistically helpful.

7. On page 6 – what is the statistical justification for having 7 stations – is there evidence that this provides sufficient data for assessment?

8. Also on page 6 the bracketed ‘(i.e chest….’) is not needed as this information has already been provided.

9. Was the training ‘based on common text’ purely written preparation? Is there any assessor standardisation?

10. I am surprised at such high scores, particularly on the CBT – is item analysis available and how is the standard set for this examination? There is very limited variance across all scores – this should be addressed as a limitation in making judgments about the correlations or lack thereof.

11. On page 11, second paragraph there is repetition – the discussion in general is repetitive and offers limited analysis of the findings. I encourage the authors to explore the potential implications more deeply to clarify a ‘so what?’ message.

12. The English grammar at the bottom of page 11 is clumsy and needs revision.

Reviewers' comments:

Reviewer's Responses to Questions

**Comments to the Author**

1. If the authors have adequately addressed your comments raised in a previous round of review and you feel that this manuscript is now acceptable for publication, you may indicate that here to bypass the “Comments to the Author” section, enter your conflict of interest statement in the “Confidential to Editor” section, and submit your "Accept" recommendation.

Reviewer #2: All comments have been addressed

2. Is the manuscript technically sound, and do the data support the conclusions?

Reviewer #2: Yes

3. Has the statistical analysis been performed appropriately and rigorously? 

Reviewer #2: Yes

4. Have the authors made all data underlying the findings in their manuscript fully available?

Reviewer #2: Yes

5. Is the manuscript presented in an intelligible fashion and written in standard English?

Reviewer #2: Yes

6. Review Comments to the Author

Reviewer #2: Comments:

Revisions well received and efforts made to address initial concerns raised but not there yet for me hence accept with minor changes.

The indepth discussions are useful for this paper to add rigour.

Minor – Typos and grammar need to be corrected.

For example -

Abstract: Although a few studies examined the efficacy of CC

Ethical consideration: rephrase the sentence – It was agreed that as all data were fully anonymised, the ethics committee waived the requirement for informed consent.

Reference 23 and 24 is a duplicate, a thorough review of the references is needed to ensure refrences in the biblography as adeqautely represented in the body of the text.

Overall - satisfied but manuscript needs to be read for grammartical and typo errors and references. Thanks.

7. PLOS authors have the option to publish the peer review history of their article (what does this mean?). If published, this will include your full peer review and any attached files.

Reviewer #2: Yes: Gozie Offiah

---

## [Author Response · Author response to Decision Letter 1]

3 Mar 2020

To the Editor (Prof. Conor Gilligan, M.D.)

Thank you for appreciating the clinical importance regarding our article.

We revised the manuscript faithfully according to the insightful comments of the editors as possible. We also deeply apologize for the insufficient revision in the previous version.

We believe that the quality of our article improved significantly by your valuable comments.

1. Editor comments: The conclusion section of the abstract needs to make a stronger ‘so what?’ point.

Thank you very much for your suggestive comments. According to your suggestion, we made the conclusion section stronger.

2. Editor comments: Reviewer 1 suggested a deeper exploration of the topic in the introduction and I do not feel that this has been addressed.

Thank you very much for your suggestive comments. As we tried to express the deeper explanation, we deeply apologize for the inconvenience regarding the introduction part in the former revision. According to your suggestion, we tried to add more explanation about the justification of this study.

3. Editor comments: On page 3 –assuring basic clinical competency is essential’ – are the authors referring to achieving this prior top CC?

Thank you very much for your suggestive comments. We apologize for the inconvenience regarding our expression. According to your suggestion, we corrected the sentence. 

4. Editor comments: On page 4, the authors have added some detail but the writing. Is unclear and includes repetition. The paragraph starting ‘The OSCE…’ needs to be revised.

Thank you very much for your suggestive comments. According to your suggestion, we reconstructed the paragraph.

5. Editor comments: Page 5 please delete ‘We have discussed that’ and begin this sentence with ‘All data’

Thank you very much for your suggestive comments. According to your suggestion, we corrected the sentence. 

6. Editor comments: Why were students who did not progress excluded? Might this have introduced bias? Also, I would expect that this group might increase the variance in the findings which would be statistically helpful. 

Thank you very much for your suggestive comments. We considered that the learning content of clinical clerkship content may differ year by year. Thus, we excluded the repeat-year students to perform correlation analysis more accurately. According to your suggestion, we added this in the limitation part.

7. Editor comments: On page 6 – what is the statistical justification for having 7 stations – is there evidence that this provides sufficient data for assessment?

Thank you very much for your suggestive comments. This is a conventional method in our OSCE evaluation. We agree with your idea. According to your suggestion, we added this in the limitation.

8. Editor comments: Also on page 6 the bracketed ‘(i.e chest….’) is not needed as this information has already been provided.

Thank you very much for your suggestive comments. According to your suggestion, we deleted the expression.

9. Editor comments: Was the training ‘based on common text’ purely written preparation? Is there any assessor standardisation?

Thank you very much for your suggestive comments. The training included standardization and also used video. According to your suggestion, we added this infomation in the method part.

10. Editor comments: I am surprised at such high scores, particularly on the CBT – is item analysis available and how is the standard set for this examination? There is very limited variance across all scores – this should be addressed as a limitation in making judgments about the correlations or lack thereof.

Thank you very much for your suggestive comments. According to your suggestion, we added the information about CBT and limitation part too. We also added information about CBT in the method part to show that quality management has been performed in this test.

11. Editor comments: On page 11, second paragraph there is repetition – the discussion in general is repetitive and offers limited analysis of the findings. I encourage the authors to explore the potential implications more deeply to clarify a ‘so what?’ message.

Thank you very much for your suggestive comments. According to your suggestion, we deleted repetition and tried to add more ‘so what’ messages and reconstructed the paragraphs. We tried to emphasize that combination of SBE and clinical environment can enhance the CC performance. We tried to express what we want to say using emergency training for clarification.

12. Editor comments: The English grammar at the bottom of page 11 is clumsy and needs revision.

Thank you very much for your suggestive comments. According to your suggestion, we corrected the English grammar as possible. 

We also performed through spell and grammar check again and corrected some parts. We also checked the references.

We sincerely appreciate for your insightful suggestions regarding our article. 

We also apologize for the inconvenience regarding our article.

We believe that our manuscript has been significantly improved by your valuable comments.

---

## [Editor Report · Decision Letter 2]

10 Mar 2020

Relationships between Objective Structured Clinical Examination, Computer-based Testing, and Clinical Clerkship Performance in Japanese Medical Students

PONE-D-19-29629R2

Dear Dr. Komasawa,

We are pleased to inform you that your manuscript has been judged scientifically suitable for publication and will be formally accepted for publication once it complies with all outstanding technical requirements.

With kind regards,

Conor Gilligan

Academic Editor

PLOS ONE

Additional Editor Comments (optional):

You have addressed the concerns adequately and I feel that the paper can now be accepted.
---

## [Editor Report · Acceptance letter]

12 Mar 2020

PONE-D-19-29629R2 

Relationships between Objective Structured Clinical Examination, Computer-based Testing, and Clinical Clerkship Performance in Japanese Medical Students 

Dear Dr. Komasawa:

I am pleased to inform you that your manuscript has been deemed suitable for publication in PLOS ONE. Congratulations! Your manuscript is now with our production department. 

With kind regards,

on behalf of

Dr. Conor Gilligan 

Academic Editor

PLOS ONE